

# Identification of key genes and crucial pathways for major depressive disorder using peripheral blood samples and chronic unpredictable mild stress rat models

Jun He[1,2], Zhenkui Ren[2,3], Wansong Xia[2], Cao Zhou[4], Bin Bi[4], Wenfeng Yu[3] and Li Zuo[1]

[1] Department of Immunology, School of Basic Medical Science, Guizhou Medical University, Guiyang, China
[2] Department of Laboratory Medicine, The Second People's Hospital of Guizhou Province, Guiyang, China
[3] Key Laboratory of Endemic and Ethnic Diseases, Ministry of Education, School of Basic Medical Science, Guizhou Medical University, Guiyang, China
[4] Psychosomatic Department, The Second People's Hospital of Guizhou Province, Guiyang, China

Corresponding author
Li Zuo, zuoligymc@163.com

## ABSTRACT

**Background**. Accurate diagnosis of major depressive disorder (MDD) remains difficult, and one of the key challenges in diagnosing MDD is the lack of reliable diagnostic biomarkers. The objective of this study was to explore gene networks and identify potential biomarkers for MDD.

**Methods**. In the present study, we performed a comprehensive analysis of the mRNA expression profiles using blood samples of four patients with MDD and four controls by RNA sequencing. Differentially expressed genes (DEGs) were screened, and functional and pathway enrichment analyses were performed using the Database for Annotation, Visualization, and Integrated Discovery. All DEGs were inputted to the STRING database to build a PPI network, and the top 10 hub genes were screened using the cytoHubba plugin of the Cytoscape software. The relative expression of 10 key genes was identified by quantitative real-time polymerase chain reaction (qRT-PCR) of blood samples from 50 MDD patients and 50 controls. Plasma levels of SQSTM1 and TNFα were measured using an enzyme-linked immunosorbent assay in blood samples of 44 MDD patients and 44 controls. A sucrose preference test was used to evaluate depression-like behavior in chronic unpredictable mild stress (CUMS) model rats. Immunofluorescence assay and western blotting were performed to study the expression of proteins in the brain samples of CUMS model rats

**Results**. We identified 247 DEGs that were closely associated with MDD. Gene ontology analyses suggested that the DEGs were mainly enriched in negative regulation of transcription by RNA polymerase II promoter, cytoplasm, and protein binding. Moreover, Kyoto Encyclopedia of Genes and Genomes pathway analysis suggested that the DEGs were significantly enriched in the MAPK signaling pathway. Ten hub genes were screened through the PPI network, and qRT-PCR assay revealed that one and six genes were downregulated and upregulated, respectively; however, SMARCA2, PPP3CB, and *RAB5C* were not detected. Pathway enrichment analysis for the 10 genes showed that the mTOR signaling pathway was also enriched. A strong positive correlation was observed between SQSTM1 and TNFα protein levels

in patients with MDD. LC3 II and SQSTM1 protein levels were increased in the CUMS rat model; however, p-mTOR protein levels were decreased. The sucrose preference values decreased in the CUMS rat model.

**Conclusions**. We identified 247 DEGs and constructed an MDD-specific network; thereafter, 10 hub genes were selected for further analysis. Our results provide novel insights into the pathogenesis of MDD. Moreover, SQSTM1, which is related to autophagy and inflammatory reactions, may play a key role in MDD. SQSTM1 may be used as a promising therapeutic target in MDD; additionally, more molecular mechanisms have been suggested that should be focused on in future *in vivo* and *in vitro* studies.

## INTRODUCTION

Major depressive disorder (MDD) is a highly prevalent mental disorder that causes suicidal death and is a significant healthcare burden worldwide (*Park & Jung, 2019*). However, there are no effective methods for the treatment and prevention of MDD. At present, diagnosis of MDD depends on the patient's self-reported symptoms and a clinician's evaluation (*Jain et al., 2016*). For this reason, differential diagnosis is difficult and often results in misdiagnosis and missed diagnosis. Therefore, it is beneficial to improve diagnosis and treatment by exploring the underlying molecular mechanisms and identifying novel biomarkers for this disorder (*Cui et al., 2016*).

Currently, the pathogenesis of MDD is still unclear, which is generally thought to be associated with complex factors, including between social pressure (*Juan et al., 2014*), genetic, epigenetic factors, and their interaction (*Kathleen et al. 2016*; *Liou et al., 2013*). One of important roles in the pathogenesis is genetic factors (*Barbu et al., 2020*). For example, 5-hydroxytryptamine (5-HT) is a very crucial monoamine neurotransmitter that regulates a variety of functions, such as locomotor activity, cognition, emotion, food intake (*Mantle et al., 1976*). 5-HT was focused upon as a key molecular in the pathogenesis of MDD (*Sa et al., 2012*). However, MDD is a common mental disorder, which is regulated by a complex gene network (*Feng et al., 2020*). Hence, screening specific gene expression profiles are promising for the use of the diagnosis and treatment of MDD.

Gene expression profiling analysis is widely used to investigate the correlations between the occurrence of diseases and differential gene expression (*Cooper-Knock et al., 2012*). Blood mRNA profiles are reportedly significantly altered and are associated with disease outcomes in diseases of the nervous system (*Laing et al., 2019*); therefore, gene expression profiling analysis is a promising method for screening disease biomarkers. Substantial efforts have been undertaken to report the abnormal expression of some genes and pathways associated with MDD. However, most of these results have only identified differentially expressed genes (*Yi et al., 2012*), and there has been an absence of further exploration of the underlying biological molecular mechanisms; this limited approach

may have inhibited the development of effective biomarkers. Thus, investigation of the molecular mechanisms of differentially expressed genes, which may allow definitive MDD diagnosis, is urgently required.

Hence, the present study evaluated potential diagnostic biomarkers and molecular mechanisms associated with MDD. Further, differential expression genes were screened between MDD patients and healthy controls. Moreover, the molecular functions of 147 differential expression genes were analyzed using gene ontology (GO) enrichment and KEGG enrichment analysis. In addition, the top 10 hub genes screened via protein–protein interaction (PPI) network were identified through 50 MDD patients and 50 healthy controls. Especially, one of the key genes was SQSTM1, which was associated with autophagy and was upregulated in the MDD patient's plasma. This finding was further confirmed in CUMS rat model. Consequently, the results of the present study indicated that SQSTM1 may play a crucial role through involved in the regulation of autophagy.

## MATERIALS AND METHODS

### mRNA isolation and RNA sequencing (RNA-Seq)

Total RNA was extracted using TRIzol® reagent (Invitrogen; Thermo Fisher Scientific, Inc.) from the blood samples of four MDD patients and four controls. We received WRITTEN informed consent from participants of our study. mRNA profiles were obtained using high-throughput next-generation sequencing, which was performed using an Illumina HiSeq 4000 platform, Genes with $p < 0.001$ were regarded as differentially expressed mRNAs (DEGs), and the gene expression was showed as the raw counts (Novogene Bioinformatics Institute, Beijing, China).

### Functional and pathway enrichment analysis

The Database for Annotation, Visualization, and Integrated Discovery (DAVID) was used to enrich the differentially expressed mRNAs (DEGs) GO and perform Kyoto Encyclopedia of Genes and Genomes (KEGG) pathway analysis. DAVID is a web tool that allows users to integrate and analyze biological data, including biological process (BP), cellular component (CC) and molecular function (MF) terms (*Da et al. 2007*). $P < 0.05$ was considered statistically significant.

### PPI network construction and module analysis

The PPI network was built by STRING, which is a useful online database for constructing a protein–protein interaction network (*Franceschini et al., 2013*). In order to evaluate their interactive associations, all DEGs were inputted to the STRING database, and chose a larger score subnetwork to further analysis through the Cytoscape software (a visual analytics platform for PPI networks). The top 10 hub genes were screened using cytoHubba plugin. The algorithms applied to identify hub genes were the maximal clique centrality (MCC) and Degree.

### Validation analysis of hub genes

The key genes were identified from the PPI network by cytoHubba plugin. Details of these key genes, such as names, abbreviations and functions were obtained by searching NCBI.

**Table 1  Primer sequences.**

| Gene | Forward Primer (5′–3′) | Reverse Primer (5′–3′) |
|------|------------------------|------------------------|
| YWHAZ | CTTGGAGGGTCGTCTCAAGT | GCTCTCTGCTTGTGAAGCAT |
| SQSTM1 | AATGGGTCCACCAGGAAACT | TTTCCCTCCGTGCTCCACAT |
| ITGB2 | ATCATGGACCCCACAAGCCT | GTCCATCAGATAGTACAGGT |
| RAB1A | CCCGGAACAGCCTATCTCAT | CTGAGTCGCCAATCAGAAGT |
| MYSM1 | AGCTAATTGGAAGCCGCACT | CATGATGGTGTCCATGCCT |
| SMARCA2 | ACTCTGAGAGAAGCTCGCAT | TTTGGAGAGCTTCTGGATCT |
| PPP3CB | AAGTAGGAGGATCACCTGCT | AAGGTGTCTGCATTCATGGT |
| AKT2 | TGTCATACGCTGCCTGCAGT | GAGCCACACTTGTAGTCCAT |
| TSC2 | ATCAAGGTGCTGGACGTGCT | TGCCAGGTCCACCAGCAACT |
| RAB5C | CTGCTGGGAACAAGATCTGT | CCCAGATCTCAAACTTGACT |
| HPRT | ATGGCGACCCGCAGCCCT | CCATGAGGAATAAACACCCT |

Subsequently, the relative expression of 10 key genes was identified using quantitative real-time PCR (qRT-PCR) of blood samples from 50 MDD patients and 50 controls. In brief, total RNA was extracted from the blood samples using TRIzol reagent (Invitrogen) following the manufacturer's instructions. StarScript II First-strand cDNA Synthesis Mix with gDNA Remover kit, which was obtained from GenStar, was used to synthesize cDNA (45 °C for 30 min and 85 °C for 5 min). SYBR-Green SuperMix (Vazyme Biotech Co., Ltd.) was used to amplify the cDNA as follows: 94 °C for 3 min, 40 cycles of 94 °C for 7 s, and 60 °C for 30 s. The relative expression of 10 key genes was calculated by the 2-$\Delta\Delta$Ct method. *HPRT* was used as an internal control. Table 1. Primer sequences.

## Plasma SQSTM1 and TNFα

SQSTM1 is involved in the regulation of autophagy and inflammation (*Chu et al., 2020*; *Yao et al., 2020*). The serum level of TNFα was increased in MDD patients, which was associated with depressive symptoms (*Alvarez-Mon et al., 2021*). We thus examined the expression of these two proteins in the plasma. Blood samples were collected from the 44 patients with MDD and 44 healthy controls. Plasma was separated from whole blood and stored at −80 °C before use. Plasma levels of total SQSTM1 and TNFα were measured using an enzyme linked immunosorbent assay (ELISA) according to the manufacturer's instructions; these two ELISA kits were purchased from Fine Test (Wuhan Fine Biotech Co., Ltd) and Elabscience (Elabscience Biotechnology Co., Ltd). Correlation analysis between SQSTM1 and TNFα was performed using the R software, version 3.5.3.

## Chronic unpredictable mild stress (CUMS) model was established

The present study was approved by the Animal Care and Welfare Committee and the Ethics Committee of Guizhou 2nd Provincial People's Hospital (permit no. 2020[06]). Twenty four female Sprague-Dawley (SD) rats weighing 180–220 g, obtained from the Experimental Animals Center at Guizhou Medical University. Rats were bred under standard conditions (12:12-h light/dark cycle, humidity 55–60%, 20–25 °C) for adaption. Twelve rats were housed in each group (six rats were used to immunofluorescence assay, six rats were used to western blotting). According to previous studies (*Yanxia et al. 2018a*; *Yanxia et*
*al., 2018b*), CUMS depression-like behavior model was established in SD rats, twelve rats underwent 21 days of CUMS procedure- induced as previously described (*Zhang et al., 2019*), with the control group being without any treatment. During a 21 days period, the CUMS group were exposed to different stimuli (food and water deprivation for 24 h, immobilization 2 h, level shaking 5 min and tail clamping). To ensure the unpredictability of the occurrence of stimulation, these stressors were changed randomly as one stressor per day in rats. Model rats were evaluated by two trained experimenters by a double blind way.

## Sucrose Preference Test (SPT)

A sucrose preference test (SPT) was used to evaluate the depression-like behavior of model rats, the procedure of SPT was performed as described previous method (*Chen et al., 2019*; *Meng-Ying et al., 2018*). First, rats in each group were trained to adapt to 1% sucrose solution (weight in volume [w/v]) with 2 bottles of 1% sucrose solution for the first day. After 24 h later, one bottle of 1% sucrose was replaced with tap water and continued adapting for 24 h. Before each test, rats were deprived of food and water for 24 h. During the 1 h SPT, rats were caged alone and, had a chance to access one bottle containing 200 mL of sucrose solution (1% w/v) and one bottle containing 200 mL of water, respectively. The positions of two bottles were randomly assigned to prevent place preference, each bottle was weighed before and after the test. According to the formula: sucrose preference =1% sucrose solution consumption/total amount of liquid intake ×100%, the sucrose preference was calculated as the percentage. Next, the rats were returned to their cages and allowed free access to food and water.

## Immunofluorescence assay

After the CUMS rat model was successfully established, the rats were euthanized and perfused transcardially with 4% paraformaldehyde. $CO_2$ inhalation was used to conduct rat euthanasia; rats were placed into enclosed cages for 5 min, $CO_2$ (100%) was then added to the cages at a displacement rate of 30% volume/min. Death was confirmed via decapitation after euthanasia after euthanasia. The rat brains were mounted on a freezing ($-80\ °C$) microtome and cut into 30 µm sections. Frozen sections were blocked with 5% normal goat serum (Sigma-Aldrich; Merck KGaA) and permeabilized with 0.3% Triton X-100 for 1 h at room temperature. Next, the slices were incubated with primary antibodies against LC3 (1:200; cat. no. 4108S; Cell Signaling Technology, Inc.) and SQSTM1 (1:200; cat. no. 23214; Cell Signaling Technology, Inc.) at 4 °C overnight. Alexa Fluor® 568-conjugated goat anti-rabbit IgG (H + L) cross-adsorbed secondary antibody (1:100; cat. no. A-11011) was purchased from Thermo Fisher Scientific and was used to bind the primary antibody for 1 h at room temperature. The slices were washed three times with PBS, before the nuclei were counterstained with DAPI (Beyotime Institute of Biotechnology) for 5 min at room temperature. Using a confocal laser scanning microscope (Olympus FV 1000; Olympus Corporation) to acquire image. Twenty randomly selected fields/sections were analyzed

## Western blotting

The brain hippocampal tissues of SD rats were collected and homogenized in RIPA buffer (1 mg/10 μL; Thermo Fisher Scientific, Inc.) containing a protease inhibitor cocktail (Sigma-Aldrich; Merck KGaA) according to a previously described method (*Wang et al., 2020*). After the lysis buffer was kept for 2 h on ice, being centrifuged at 12,000 rpm for 20 min. The protein contents were determined by a bicinchoninic acid assay kit (cat. no. P0009; Beyotime Institute of Biotechnology), next, 25 μg proteins were separated by 12% SDS-PAGE and transferred onto nitrocellulose membranes. 5% nonfat milk was used to block the membrane at room temperature for 2 h, before incubated with primary antibodies against the autophagy-related proteins from Cell Signaling Technology company: phosphorylated (p)-mTOR (1:1,000; cat. no. 5536S), mTOR (1:1,000; cat. no. 2972S), SQSTM1 (1:1,000; cat. no. 23214), light chain (LC)3II/I (1:1,000; cat. no. 4108S), β-actin (1:1,000; cat. no. 3700S), and TNFα (1:1,000; cat. no. ab205587) from Abcam overnight at 4 °C. Subsequently, a horseradish peroxidase-conjugated secondary antibody (1:1,000; cat. no. 7074S; Cell Signaling Technology, Inc.) was used to incubate membranes at room temperature for 2 h, and the membranes were visualized by an enhanced chemiluminescence reagent kit (Amersham; Cytiva). Gray values of proteins were measured using ImageJ software (version 1.42q, National Institutes of Health) and normalized to that of β-actin, which was the internal control.

## Statistical analysis

All statistical analyses were conducted using GraphPad Prism software 7.0 (GraphPad Software, Inc.), and data are shown as mean ± standard deviation. Multiple groups comparison analysis was performed using one-way analysis of variance followed by Tukey's post hoc test; comparisons were made between the two groups through unpaired t-tests. $P < 0.05$ was considered a statistically significant difference.

# RESULTS

## Differentially expressed genes identification

RNA-seq analysis between the MDD and control blood samples revealed the DEGs (16272 genes Genes detected in this transcriptome study). We drew a heat map using four biological replicates for each group (Fig. 1A). The volcano plot showed that 180 mRNAs were significantly upregulated and 67 mRNAs were significantly downregulated according to *P*-value <0.001 (Fig. 1B).

## Functional and pathway enrichment analysis of DEGs

To explore biological relevance of the 247 DEGs by DAVID database. GO analysis included BP, MF, and CC. Our results suggested that the DEGs were mainly enriched in negative regulation of transcription from RNA polymerase II promoter, in the cytoplasm, and in protein binding (Fig. 2A). Moreover, KEGG pathway analysis indicated that the DEGs were significantly enriched in the MAPK signaling pathway (Fig. 2B).
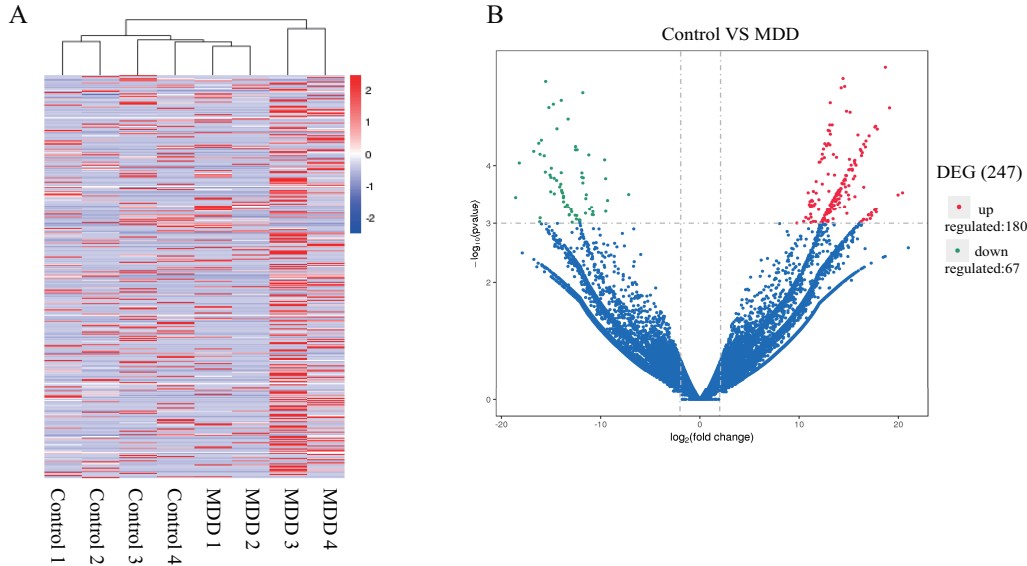

Figure 1 **Screening of differentially expressed genes.** (A) Heatmap showing the differential expression of mRNA between four MDD patients and four healthy controls. Highly expressed mRNAs are showed in red and lowly expressed mRNAs in green. (B) Differentially expressed genes are represented as a Volcano plot (*P*-value < 0.001).

## PPI network analysis of DEGs

The interactions of 247 DEGs were explored using the STRING online database to investigate the PPI network likely associated with MDD. The obtained PPI network were visualized by the Cytoscape software (Fig. 3A). The top 10 hub genes were screened using the cytoHubba plugin (Fig. 3B). To better understand the functions of 10 hub genes, there is a need to perform a more systematic analysis through pathway enrichment in DAVID database (Fig. 3C). The following top 10 hub genes were identified from the PPI network using the cytoHubba plugin: *YWHAZ, SQSTM1, ITGB2, RAB1A, MYSM1, SMARCA2, PPP3CB, AKT2, TSC2*, and *RAB5C*. Details of these key genes, such as names, abbreviations, and functions, are provided in Table 2.

## Validation analysis of the 10 hub genes

The mRNA expression of 10 hub genes were identified between the MDD and control blood samples by q-PCR. A total of 1 decreased gene and 6 increased genes were detected, however, *SMARCA2, PPP3CB* and *RAB5C* were not detected (Fig. 4).

## Positive correlation between the levels of SQSTM1 and TNFα

Our results showed that the protein levels of SQSTM1 and TNFα were significantly increased in patients with MDD (Figs. 5A, 5B). Further analysis revealed a strong positive correlation was seen between SQSTM1 and TNFα protein levels (Fig. 5C).

## Autophagy related proteins were detected in CUMS rat models

SQSTM1 mRNA level is higher in MDD blood samples than control blood samples. To further explore SQSTM1 protein level changes in the CUMS rat models, it is important

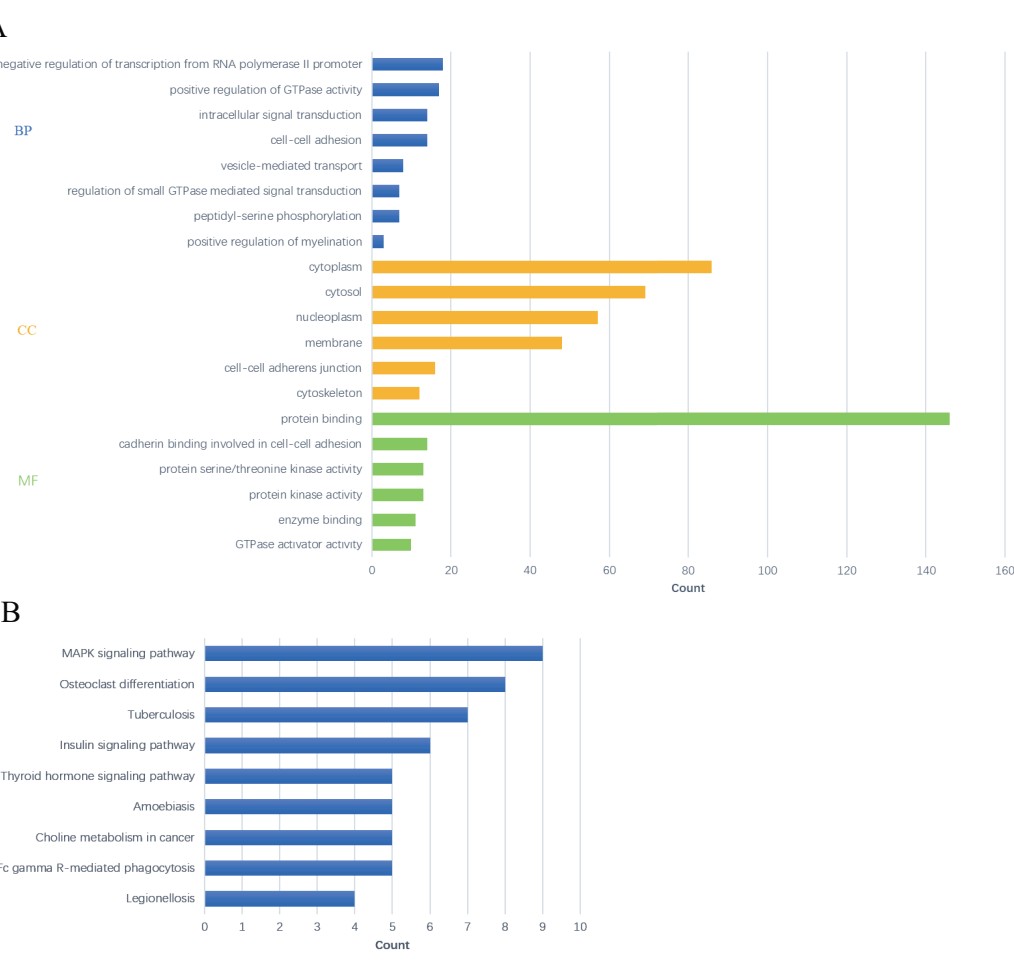

**Figure 2  GO and KEGG enrichment analyses of differentially expressed genes (DEGs) between the control blood samples and the MDD blood samples.** (A) GO Functional Enrichment Map of DEGs; blue represented biological process; brown represented cellular component; green represented molecular function (Count; $p$-value $< 0.05$). (B) The first nine significantly enriched pathways of 247 DEGs were shown by their scores (Count; $p$-value $< 0.05$).

to identify the specific function of SQSTM1. The *in vivo* results from the present study demonstrated that SQSTM1 protein levels were increased in the CUMS rat models (Figs. 6A, 6D). Meanwhile, LC3, which is an autophagy activated marker, was upregulated in the CUMS rat model (Figs. 6B, 6C). mTOR a key signaling pathway associated with negatively regulating autophagy, was inhibited (Fig. 6E). SPT was used to evaluate depression-like behavior in the model rats (Fig. 6F).

## DISCUSSION

In the first part of our study, we screened 247 DEGs between the blood samples of four MDD patients and four controls. Based on the BP annotations, negative regulation of transcription by RNA polymerase II promoter, positive regulation of GTPase activity, intracellular signal transduction and cell–cell adhesion were suggested to be significantly

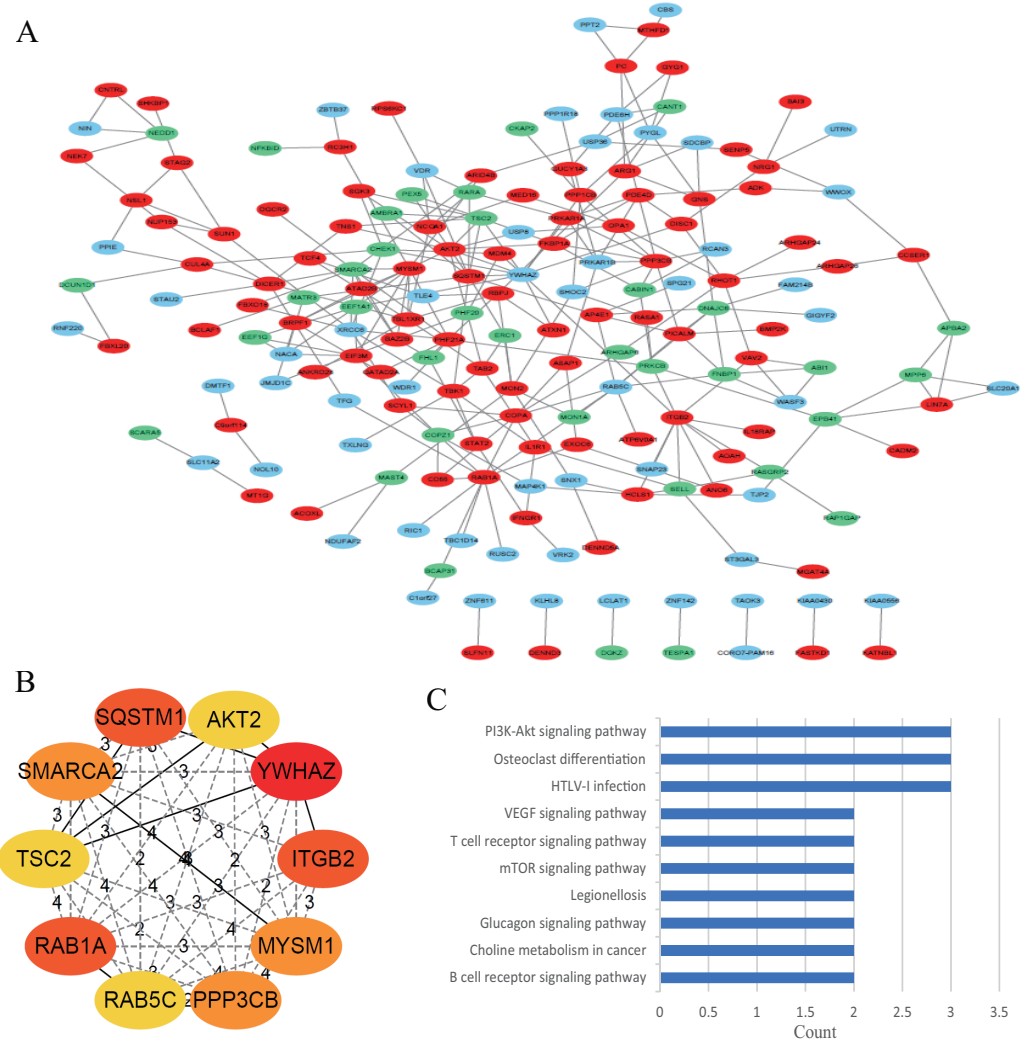

**Figure 3** **Protein–protein interaction (PPI) network construction and analysis of the most significant module of differentially expressed genes (DEGs).** (A) The PPI network of DEGs was constructed in STRING database (https://string-db.org/) and visualized in Cytoscape. Highly expressed mRNAs are showed in red and lowly expressed mRNAs in green, the unchanged mRNAs in blue. (B) The most significant module was obtained from the PPI network by using cytoHubba plugin. (C) The first 10 significantly enriched pathways of 10 hub genes are shown by their scores (Count; *P*-value < 0.05).

involved in the pathologic processes of MDD. Of the CC annotations, the cytoplasm, cytosol, nucleoplasm and membrane were found to play important roles in MDD. According to the MF annotations, protein binding, cadherin binding involved in cell–cell adhesion, protein serine/threonine kinase activity and protein kinase activity were predicted to be significantly correlated with the occurrence and development of MDD. Moreover, KEGG pathway analysis indicated that the DEGs were significantly enriched in the MAPK signaling pathway. MAPK pathways are involved in the induction of inflammatory factors (*Inamdar et al., 2014*). Further research found that TNFα is also upregulated in plasma of

**Table 2  Functional roles of 10 hub genes.**

| NO. | Gene Name | Description | Function |
|---|---|---|---|
| 1 | YWHAZ | tyrosine 3-monooxygenase/tryptophan 5-monooxygenase activation protein zeta | Regulates spine maturation through the modulation of ARHGEF7 activity |
| 2 | SQSTM1 | sequestosome 1 | Autophagy receptor required for selective macroautophagy (aggrephagy), May regulate the activation of NFKB1 by TNF-alpha, nerve growth factor (NGF) and interleukin-1. |
| 3 | ITGB2 | integrin subunit beta 2 | Integrin ITGAL/ITGB2 in association with ICAM3, contributes to apoptotic neutrophil phagocytosis by macrophages |
| 4 | RAB1A | RAB1A, member RAS oncogene family | RAB1A regulates vesicular protein transport from the endoplasmic reticulum (ER) to the Golgi compartment and on to the cell surface,and plays a role in IL-8 and growth hormone secretion |
| 5 | MYSM1 | Myb like, SWIRM and MPN domains 1 | Acts as a coactivator by participating in the initiation and elongation steps of androgen receptor (AR)-induced gene activation |
| 6 | SMARCA2 | SWI/SNF related, matrix associated, actin dependent regulator of chromatin, subfamily a, member 2 | Involved in transcriptional activation and repression of select genes by chromatin remodeling (alteration of DNA-nucleosome topology) |
| 7 | PPP3CB | protein phosphatase 3 catalytic subunit beta | Dephosphorylates and inactivates transcription factor ELK1 |
| 8 | AKT2 | AKT serine/threonine kinase 2 | AKT2 regulates many processes including metabolism, proliferation, cell survival, growth and angiogenesis. |
| 9 | TSC2 | TSC complex subunit 2 | A direct activator of the protein kinase activity of mTORC1 |
| 10 | RAB5C | RAB5C, member RAS oncogene family | RAB5C, Ras oncogene superfamily member,involved in endocytosis |

MDD, which is in agreement with a previous study (*Li et al., 2012*). Taken together, these findings shed light on the inflammatory factors that may be major contributors to MDD.

In the second part of our study, in order to evaluate their interactive associations, all DEGs were inputted to the STRING database to build the PPI network, and a subnetwork with a large score was chosen for further analysis using Cytoscape software. The top 10 hub genes were screened using the cytoHubba plugin. Further investigation of these key genes may reveal novel molecular mechanisms underlying MDD. Next, the mRNA expression of 10 hub genes was assessed in the blood samples of 50 MDD patients and 50 controls by qRT-PCR. One downregulated gene and six upregulated genes were detected; however, SMARCA2, PPP3CB and RAB5C were not detected. Uncovering the molecular functions of SQSTM1 deserved more attention in our study. SQSTM1 is a biomarker of autophagy, and is predicted to be associated with inflammatory factors (*Lee et al., 2011*). Moreover, the levels of SQSTM1 and TNFα were positively correlated based on the ELISA. Therefore, we speculated that SQSTM1 may be associated with the regulation of inflammatory reactions. Inflammation is supposed to play an important role in the pathophysiology of MDD (*Marazziti et al., 2020*).

To expore further details of the 10 hub genes, their functions were searched using NCBI. Our results showed that YWHAZ regulates spine maturation through the modulation of

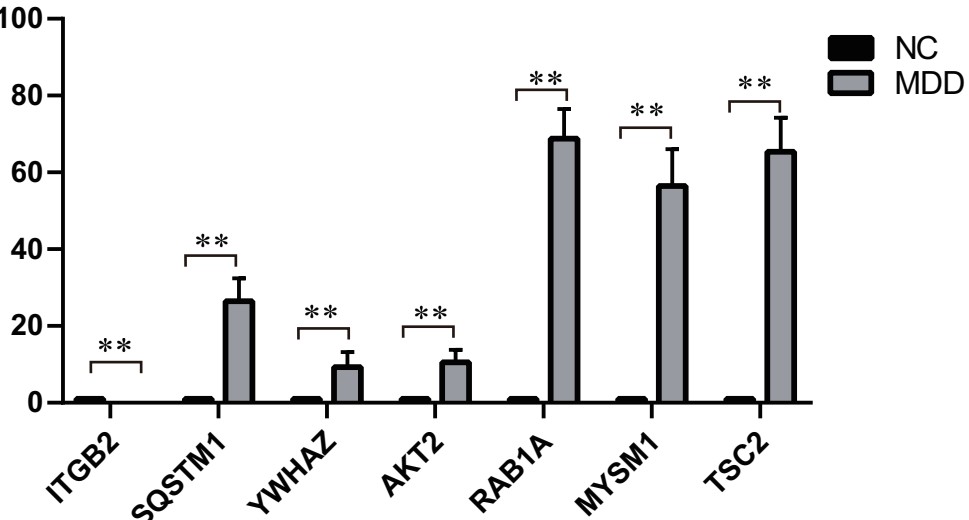

**Figure 4** Quantitative real-time PCR analysis of the differences in expression of mRNA of 10 hub genes between controls and patients with major depressive disorder. A total of one downregulated gene (*ITGB2*) and six upregulated genes (*YWHAZ, SQSTM1, RAB1A, MYSM1, AKT2, TSC2*) were detected; however, *SMARCA2, PPP3CB* and *RAB5C* were not detected. ** $P < 0.01$.

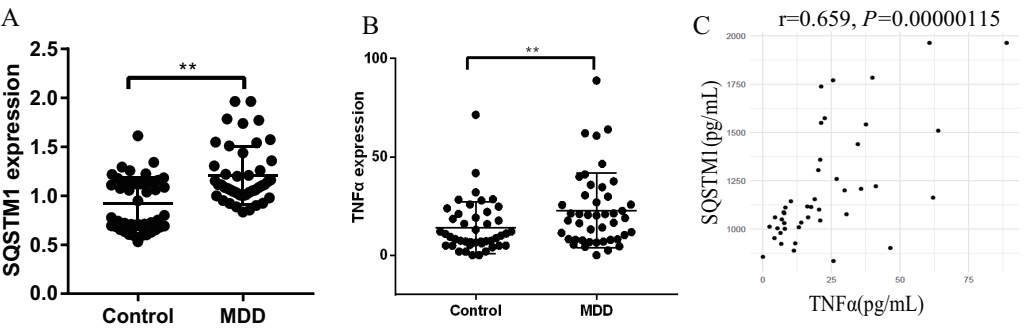

**Figure 5** Quantification of SQSTM1 and TNFα levels in plasma. (A) The protein level of SQSTM1 was increased in the plasma (ng/mL). (B) The protein level of TNFα was upregulated in the plasma (pg/mL). (C) A strong positive correlation was showed between SQSTM1 and TNFα protein ($r = 0.659$, $P = 0.00000115$). Data are presented as mean ± standard deviation; n =44/group. ** $P < 0.01$.

ARHGEF7 activity, and could be associated with depression in patients with Alzheimer's Disease (AD) (*Yang et al., 2020*). SQSTM1 might regulate the activation of NFKB1 by TNF-alpha, and the overexpression of SQSTM1 could ameliorate spatial learning and long-term memory, indicating that SQSTM1 is a potential diagnostic marker and therapeutic target. RAB1A is involved in the inflammatory signaling pathway (*Song et al., 2020*). Akt2 knockout led to more significantly anxiety and depressive behaviors than corresponding wild type mice; therefore, this might be a crucial factor in the pathophysiology of depression and anxiety (*Leibrock et al., 2013*). TSC2 mutations are related to the level of long-term depression via regulation of hippocampal synapse function (*von der Brelie et al., 2006*).

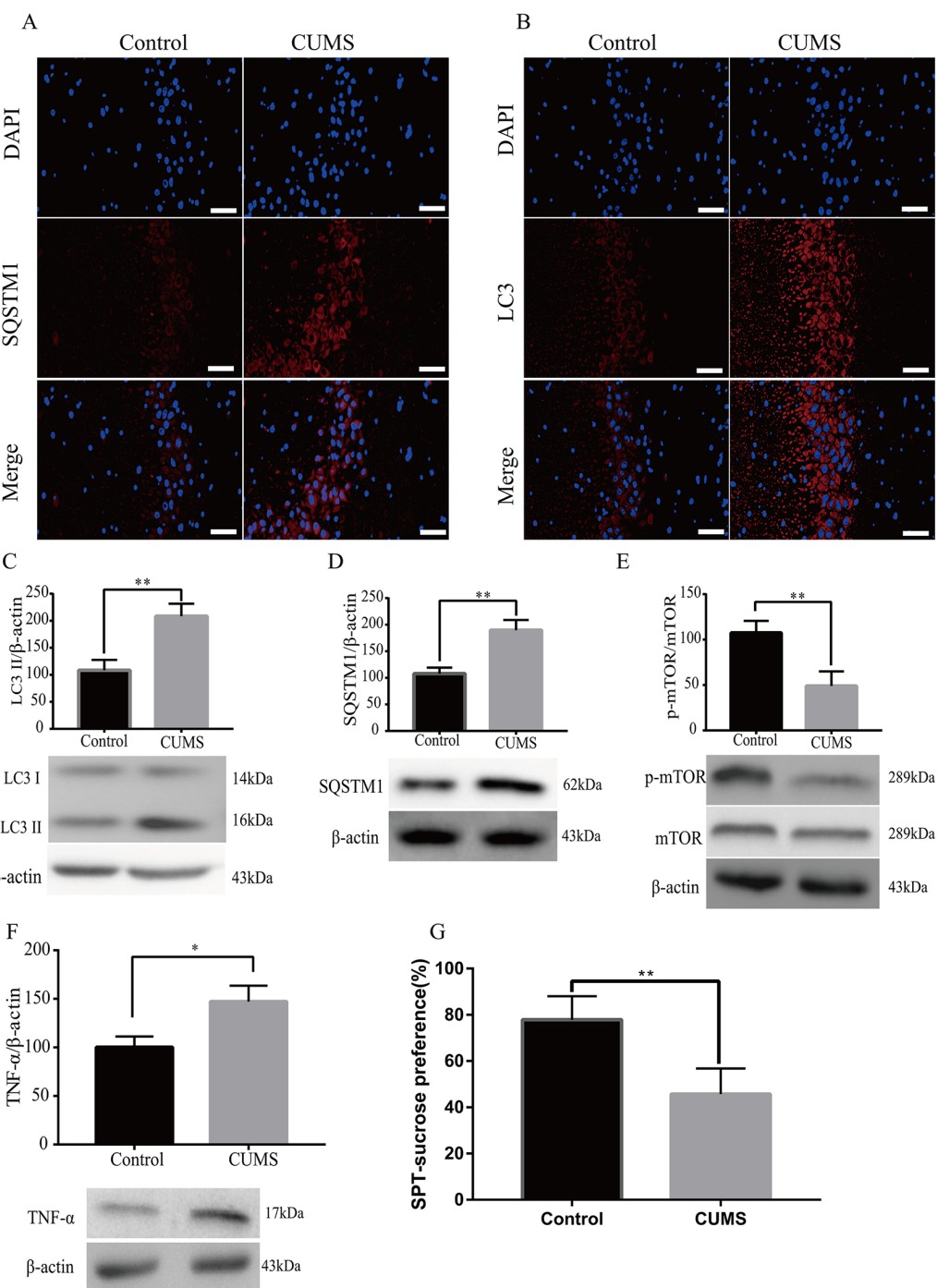

**Figure 6 Autophagy related proteins were detected in CUMS rat models.** Immunofluorescence double-labeling showed the upregulation of SQSTM1 (A) and LC3 (B) in CUMS rat models. (C, D) LC3 II and SQSTM1 protein levels were increased in CUMS rat models according to the western blot assay. (E) p-mTOR protein level was decreased in CUMS rat model as per the western blot assay. (F) TNFα protein levels were increased in CUMS rat models according to the western blot assay. (G) The sucrose preference values were decreased in CUMS rat models. Scale bar = 50 μm. Data are presented as mean ± standard deviation; n =6/group. * $P < 0.05$, ** $P < 0.01$.

The molecular mechanism of above five hub genes involved in MDD deserves further investigation. However, the relationship between the remaining five hub genes (MYSM1, ITGB2, SMARCA2, RAB5C, PPP3CB) and MDD have not been reported, it could be that these five hub genes could have potential function in MDD.

For a better understand the roles of 10 hub genes, there is a need to perform a more systematic analysis through pathway enrichment in DAVID database. In particular, we found that the mTOR, which is involved in the regulation of autophagy and inflammation signaling pathway (Chu et al., 2020; Yao et al., 2020) was enriched. Further study found that mTOR signaling pathway is inhibited in CUMS rat model, which is in agreement with the results reported previously (Zhang et al., 2020; Zhuo et al., 2020). SQSTM1, which was also upregulated in CUMS rat model, is associated with autophagy. Moreover, the expression of LC3II was increased in the CUMS rat model, which is consistent with the results reported previously (Zhang et al., 2020). Therefore, it is reasonable to speculate that SQSTM1 could play a crucial role through regulated autophagy in MDD.

## CONCLUSIONS

In conclusion, this study was to explore the molecular function of DEGs via a comprehensive bioinformatics analysis, and to identify hub genes were involved in the development of MDD. Especially, the expression of 10 hub genes was confirmed between 50 MDD patients and 50 healthy controls, however, SMARCA2, PPP3CB and RAB5C were not detected. SQSTM1 was significantly focused on in the present study, which was associated with autophagy and inflammatory reactions. Further analysis revealed a strong positive correlation was seen between SQSTM1 and TNFα protein levels in the plasma of MDD. SQSTM1 may be used as a promising therapeutic target in MDD; additionally, several other molecular mechanisms have been suggested, which should be focused on in future *in vivo* and *in vitro* studies.

### Funding
This work was supported by the National Natural Science Foundation of China (grant no. 81360199). The funders had no role in study design, data collection and analysis, decision to publish, or preparation of the manuscript.

### Grant Disclosures
The following grant information was disclosed by the authors:
The National Natural Science Foundation of China: 81360199.

### Competing Interests
The authors declare there are no competing interests.

### Author Contributions
- Jun He conceived and designed the experiments, prepared figures and/or tables, and approved the final draft.

- Zhenkui Ren and Wansong Xia performed the experiments, prepared figures and/or tables, and approved the final draft.
- Cao Zhou, Bin Bi and Wenfeng Yu analyzed the data, authored or reviewed drafts of the paper, and approved the final draft.
- Li Zuo conceived and designed the experiments, authored or reviewed drafts of the paper, and approved the final draft.

## Human Ethics

The following information was supplied relating to ethical approvals (i.e., approving body and any reference numbers):

The Ethics Committee of the Second People's Hospital of Guizhou Province granted Ethical approval to carry out the study within its facilities [Ethical Application Ref: 2020 (06)].

## Animal Ethics

The following information was supplied relating to ethical approvals (i.e., approving body and any reference numbers):

The Animal Care Welfare Committee of the Second People's Hospital of Guizhou Province granted Ethical approval to carry out the study within its facilities [Ethical Application Ref: 2020 (06)].

## DNA Deposition

The following information was supplied regarding the deposition of DNA sequences:

The sequences in the Supplemental File are available at NCBI Gene.

The raw sequencing data is available at NCBI: PRJNA738649.

## Data Availability

The raw measurements are available in the Supplemental Files.

## Supplemental Information

Supplemental information for this article can be found online at http://dx.doi.org/10.7717/peerj.11694#supplemental-information.

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
