# Peer review of "Identification of key genes and crucial pathways for major depressive disorder using peripheral blood samples and chronic unpredictable mild stress rat models"

_PeerJ, doi:10.7717/peerj.11694_

## Round 0.1 · original submission · Major Revisions

As stated by the reviewers, some conclusions are not fully supported by the results. The study would be improved by additional experiments and data re-analysis.

·

Basic reporting

This is an interesting study, and much work have been done to elucidate this topic.
While there are still much major and minor concerns for this study and manuscript.
Minor concerns:
1. Line 86, "This finding further was confirmed in CUMS rat model" was repeatedly.

2. Line 136, the last word should be "Guizhou" rather than "Guizbou".

Experimental design

1. Why in the preclinical study, the female rats were selected as the animal model? how can author exclude the bias because of the menstrual cycle, especially in the female adults rats?

2. And totally 7 proteins were found differently between patients with MDD and healthy control, why author selected only two of them for the further study?

3. Why author selected the brain hippocampal tissues of SD rats to perform the Western blotting? How about the mPFC or other brain regions?

Validity of the findings

Author should add the examination of the other 5 proteins in the animal model, MYSM1, ITGB2, SMARCA2, RAB5C and PPP3CB. And Provide more evidences to support their discussion, for example, the 3 and 4 paragraph in the discussion.

Additional comments

1. Line 86, "This finding further was confirmed in CUMS rat model" was repeatedly.

2. Line 136, the last word should be "Guizhou" rather than "Guizbou".

3. Why in the preclinical study, the female rats were selected as the animal model? how can author exclude the bias because of the menstrual cycle, especially in the female adults rats?

4. And totally 7 proteins were found differently between patients with MDD and healthy control, why author selected only two of them for the further study?

5. Why author selected the brain hippocampal tissues of SD rats to perform the Western blotting? How about the mPFC or other brain regions?

Reviewer 2 ·

Basic reporting

The paper was not well written with some details missing.
The overall results were weak.
The raw data were shared in public datasets.

Experimental design

For the RNASeq, the sample size is too small. Further, the lack of basic sample evaluations made the further judgment of their work difficult.

Validity of the findings

Weak connections between the validation rats and their RNASeq results.

Additional comments

Review of Manuscript 56643, " Identification of key genes and crucial pathways for major depressive disorder using peripheral blood samples and chronic unpredictable mild stress rat models"

This paper analyzed the RNAseq data of blood samples from a total of 8 unique individuals, including 4MDD and 4 controls. A total of 247 DEGs, including 180 upgraded and 67 downgraded, were identified. These DEGs were annotated using GO and KEGG genesets. In addition, the DEGs were also put to String-db for network analysis. Using cytoscape cytoHubba plugin, 10 hub genes were identified and further annotated using David online annotation tools. Expression of these 10 hub genes was verified using qRT-PCR of blood samples from 50 MDD and 50 controls. Among these 10 genes, SQSTM1 was further studied and the protein level of SQSTM1 was shown to be positively correlated with TNFa, an inflammation marker with a sample size of 44 MDD and 44 controls. Finally, in the brain, SQSTM1 were also found to be overexpressed in CUMS rats.
Before publications, there are still a lot of questions that need to be addressed. The most important results of this paper came from DEG analysis of the blood of human subjects, with only 4 samples per group. Given the high diversity of human subjects, these results were not reliable especially that the authors wanted to use the gene expressions in the blood to study a brain disorder.
Besides, there are some other technical details that need to be clarified.

Main comments:
1. Please add a table describing the basic demographic information of the key 8 human subjects. This table should include the information of sex, age(or age range), medicines, etc. Since only 4 subjects per group, the review wanted to exclude the possibilities of the effect from other covariables.
2. From Figure 1A, it seems that their results were driven from MDD3. A PCA plot of the 8 subjects may help to check whether MDD3 is an outlier or not.
3. In Figure 1, the log2fold change of most DEGs were above 10. This means a 1000 fold increase. This is unlikely based on the heatmap in A. Please check. Also, in 1A, please add a legend of the values in the heatmap. If it was the gene expression, please add whether it is the raw counts or lcpm, or something else. In the related method part, please add details on how the sequencing data were processed.
4. In Figure 2 and Figure 3C, Please add p-value information.
5. Related to Figure 3B, the details of how the 10 hubs were selected should be described, including the rules of the thresholds, and the basic algorithms of the plugin.
6. Colors in Figure 3A is not described. Also, in String-db, the thresholds of the chosen PPI were not described.
7. Why only SQSTM1 were selected for further studies?

Reviewer 3 ·

Basic reporting

Overall, the paper is written in good format with minor editing in the languages. The reference are properly used.

Experimental design

The experimental design has sufficient amount of biological replicates and the data has been analyzed and validated using various techniques. There are some gaps that need to be filled. Please find the details in the attached documents.

Validity of the findings

The authors need to provide the raw files for the sequencing data. The fold change in the sequencing data needs to be shown and along with some background about the clinical data to appreciate the differences seen in figure 1.

Additional comments

Overall it is well thought out plan. The authors need to address some of the comments to strength the manuscript.

The authors have a good number of samples for their RNA sequencing data, validation, and rat model data. However, there are some blanks that need to be filled to strength the findings.
• TNF-α needs to be questioned in the rat model (either using IF or western blot)
• Need to validate the inflammatory response caused by TNF-α using the samples from rat model or patients.
• Role of SQSTM1 in other mental health disorders has been established.
• Some of the other genes such as AKT2, TSC2, RAB5C, ITGB2 are involved in autophagy and phagocytosis need to be validated in the rat model or in the human samples using rtPCR to strengthen the manuscript.
• There are minor edits that needs to be addressed to the text

Annotated reviews are not available for download in order to protect the identity of reviewers who chose to remain anonymous.

---

## Round 0.2 · Minor Revisions

The reviewers are of the opinion that your paper has been significantly improved. However, there are some concerns that must be addressed. Please see the comments of reviewer 2.

Reviewer 2 ·

Basic reporting

In the revised manuscript and response letter, the authors addressed most of my questions. The remaining concern is about the data availability policy, which would make it less efficient for repeats and comparisons.
Something minor things still need to be addressed:
1.The original question 6: Still need to provide detailed descriptions about the cut-offs of your PPI. The score cut-off is to limit the number of interactions. Setting different cutoffs will affect your final results.
2." The algorithms applied to identify hub genes were MCC and Degree", what is "MCC" short for? Could the author briefly describe the algorithms in the revised version?

Experimental design

no comment

Validity of the findings

no comment

Reviewer 3 ·

Basic reporting

The authors written the manuscript well and included all the raw data

Experimental design

The experiment design is accurate to test and validate the hypothese

Validity of the findings

The findings have been validated by various techniques which has helped strengthen the data

---

## Round 0.3 · accepted · Accept

Thanks for your effort in improving the manuscript.

Reviewer 2 ·

Basic reporting

In the revised version and the response letter, the author addressed my questions.

Experimental design

no comment

Validity of the findings

no comment